# Damage to the right insula disrupts the perception of affective touch

Louise P Kirsch[1,2]*, Sahba Besharati[3], Christina Papadaki[1], Laura Crucianelli[1,4], Sara Bertagnoli[5], Nick Ward[6], Valentina Moro[5], Paul M Jenkinson[7], Aikaterini Fotopoulou[1]

[1]Department of Clinical, Educational and Health Psychology, University College London, London, United Kingdom; [2]Institut des Systèmes Intelligents et de Robotique, Sorbonne Université, Paris, France; [3]Department of Psychology, University of the Witwatersrand, Johannesburg, South Africa; [4]Department of Neuroscience, Karolinska Institutet, Stockholm, Sweden; [5]NPSY.Lab-VR, Department of Human Sciences, University of Verona, Verona, Italy; [6]Sobell Department of Motor Neuroscience and Movement Disorders, UCL Institute of Neurology, London, United Kingdom; [7]Department of Psychology, School of Life and Medical Sciences, University of Hertfordshire, Hatfield, United Kingdom

**Abstract** Specific, peripheral C-tactile afferents contribute to the perception of tactile pleasure, but the brain areas involved in their processing remain debated. We report the first human lesion study on the perception of C-tactile touch in right hemisphere stroke patients (N = 59), revealing that right posterior and anterior insula lesions reduce tactile, contralateral and ipsilateral pleasantness sensitivity, respectively. These findings corroborate previous imaging studies regarding the role of the posterior insula in the perception of affective touch. However, our findings about the crucial role of the anterior insula for ipsilateral affective touch perception open new avenues of enquiry regarding the cortical organization of this tactile system.

*For correspondence:
kirsch.lou@gmail.com

Competing interests: The authors declare that no competing interests exist.

## Introduction

Increasing evidence points to the importance of affective touch to human development and health (*McGlone et al., 2014*). It has been proposed that humans, like other mammals, have a specialized neurophysiological system for tactile affectivity (in particular, pleasant sensations arising from the skin; called the 'CT system', *McGlone et al., 2014*; *Croy et al., 2016*), separate from that for touch discrimination (*Vallbo et al., 1999*; *Essick et al., 1999*; *Olausson et al., 2002*). Specifically, in the peripheral nervous system, affectivity of touch has been linked to the activation of unmyelinated, mechanosensitive C-tactile fibers (CTs) that are present only in hairy skin and respond preferentially to low pressure, slow stroking touch at skin temperature (*Löken et al., 2009*; *Ackerley et al., 2014*), in opposition to fast conducting myelinated (Aβ) fibers that provide the brain fast sensory information about tactile stimulations, including their duration, texture, force, velocity and vibration (*Johansson and Vallbo, 1979*; *Vallbo and Johansson, 1984*). Microneurography studies found that CTs are velocity tuned, responding optimally to a stimulus moving over their receptive field at between 1 and 10 cm/s, with discharge frequencies that correlate with subjective ratings of stimulus pleasantness as measured psychophysically (*Vallbo and Hagbarth, 1968*; *Nordin, 1990*; *Vallbo et al., 1999*; *Löken et al., 2009*). Functional neuroimaging studies have demonstrated a functional segregation, with primary and secondary somatosensory cortices most commonly associated with discriminatory touch (Aβ mediated), while tactile pleasantness (CT mediated) is associated with other areas such as the posterior insula (*Björnsdotter et al., 2009*; *McGlone et al., 2012*; *Morrison, 2016*), parietal operculum, orbitofrontal cortex and superior temporal sulcus (*Perini et al.,*

2015; *Björnsdotter, 2016*). C-tactile afferents have been shown to take a distinct ascending pathway from the periphery to the posterior insula (*Olausson et al., 2002*; *Morrison et al., 2011*, but see also *Marshall et al., 2019*), which is understood to support an early convergence of sensory and affective signals about the body that are then re-represented in the mid- and anterior insula, the proposed sites of integration of interoceptive information with other contextual information (*Critchley et al., 2004*; *Craig, 2009*; *Evrard and Craig, 2015*). However, these studies are correlational. Only two neuromodulatory, repetitive transcranial magnetic stimulation (rTMS) studies (*Case et al., 2016*; *Case et al., 2017*) have investigated causal relationships, finding that the right primary and secondary somatosensory cortex are not necessary for the perceived affectivity of touch. The causative role of the insular cortex, subcortical structures and white matter connections has not yet been studied, as virtual lesion methods have limited validity when applied to these deeper regions (*Lenoir et al., 2018*). By contrast, lesion studies allow for direct examination of the functional role of both superficial and deep brain areas.

Accordingly, we aimed to investigate for the first time the right hemisphere regions which are necessary for the perceived affectivity of CT-optimal touch, applying a voxel-based lesion symptom mapping approach (VLSM; *Bates et al., 2003*) in a large, consecutively recruited cohort of patients (N = 59) with recent, first-ever, right hemisphere lesions following a stroke. Contrary to other neuropsychological approaches that employ diagnostic, group comparisons, the VLSM method uses continuous measures in a single sample, and identifies which regions of the brain are crucial to a specific behavior (e.g. here CT pleasantness perception), without assuming that all patients show the same tactile profile. The selection of right-hemisphere patients restricts any laterality interpretations, but it also avoids the possibly confounding sequelae of left hemisphere lesions, such as language and depression symptoms (*Robinson et al., 1984*; *Whyte and Mulsant, 2002*).

We used a previously validated tactile stimulation paradigm (*Crucianelli et al., 2013*; *Crucianelli et al., 2018*; *Gentsch et al., 2015*; *Mohr et al., 2017*; *Kirsch et al., 2018*), together with standardized neuropsychological, somatosensory and mood assessments. Our affective touch paradigm required blindfolded patients (N = 59, RH) and age-matched healthy controls (N = 20, HC), to rate the intensity and pleasantness of brushing stimuli delivered at velocities known to activate the CT-system optimally (3 cm/s; CT-optimal affective touch) or not (18 cm/s; CT-suboptimal neutral touch) to both the left (contralesional) and the right (ipsilesional) forearm (*Vallbo et al., 1999*; *Löken et al., 2009*). This touch on the forearm stimulates both Aβ and CT fibers; one cannot stimulate one type of fiber without stimulating the other simultaneously (except in patients without Aβ afferents, as studied by *Olausson et al., 2002*; *Olausson et al., 2008*). However, our paradigm is optimized to stimulate CT fibers differentially based on velocity, and the resulting difference in pleasantness, that is CT pleasantness sensitivity, is assumed to be at least partly linked to the differential involvement of these CT fibers (even if not restricted to it). Specifically, Aβ fiber activation is known to linearly increase with increases in velocity, while the mean frequency firing rate of CT fibers follows an inverted U shape with higher firing being in the 1–10 cm/s range, and have been shown to be the only unit types for which firing patterns correlate with average psychophysiological ratings, that is pleasantness (*Löken et al., 2009*). In addition to the affective touch paradigm, to control for general pleasantness deficits (specific to touch), participants had to imagine being touched by pleasant (i.e. velvet) and unpleasant (i.e. sandpaper) materials and rate the associated pleasantness.

Given that right hemisphere and particularly right perisylvian regions have been previously associated with somatosensory and interoceptive perception (*Dijkerman and de Haan, 2007*; *Preusser et al., 2015*), we expected our patients to have, on average, reduced ratings of both touch intensity and pleasantness in comparison to healthy controls, and particularly in the contralesional left forearm. An overall reduced tactile pleasantness in patients (both in actual touch and imagined touch pleasantness ratings) would suggest tactile anhedonia linked to general right hemisphere lesions. Crucially, given the assumed neurophysiological specificity of the CT system, we expected that more specific lesions to the posterior insula (*Morrison, 2016*) would reduce the affective sensitivity of these patients to CT-optimal touch, over and above general effects of anhedonia, tactile acuity and other neuropsychological deficits caused by the broader lesion profile of our whole sample. In other words, an intact posterior insula should be necessary for the added affective sensitivity that the CT fibers are conveying during touch optimally activating the CT system versus an identical touch and social context that does not activate this afferent pathway optimally. Moreover, this would give further substance to the hypothesis that the CT afferent pathway is a specialized system that

## Results and discussion

In the present study, we used a previously validated affective touch protocol in stroke patients to investigate, for the first time, the right hemisphere regions which are necessary for the perceived affectivity of CT-optimal touch, applying a voxel-based lesion symptom mapping approach.

First, we investigated the effect of right hemisphere lesions on the perception of touch intensity and pleasantness, on the contralesional and ipsilesional forearm separately, by comparing stroke patients' and healthy controls' intensity and pleasantness ratings in turn. In line with the high percentage of contralesional tactile deficits in right hemisphere stroke patients (including in our patients' sample, see Materials and methods), patients, as compared to healthy controls, perceived touch, regardless of velocity, as less intense on the contralesional forearm (contralesional: $F_{(1,57)}$=55.918, p<0.001, $\eta_p^2$=.495; $BF_{10}$ = 1.480*$10^7$; ipsilesional: $F_{(1,38)}$=0.834, p=0.367, $\eta_p^2$ = .021, $BF_{10}$ = 0.759; see *Figure 1A and B*). Most interestingly, we observed a main effect of stroking type on pleasantness ratings, with both patients and controls rating CT-optimal affective touch as more pleasant than CT-suboptimal neutral touch on both forearms (contralesional: $F_{(1,53)}$=22.444, p<0.001, $\eta_p^2$ = .297, $BF_{10}$ = 3526.340; ipsilesional: $F_{(1,59)}$=11.519, p=0.001, $\eta_p^2$ = .163, $BF_{10}$ = 38.833; *Figure 1C and D*). Moreover, patients perceived touch as less pleasant than controls on both forearms (contralesional: $F_{(1,53)}$=14.074, p<0.001, $\eta_p^2$=.210, $BF_{10}$ = 62.636; ipsilesional: $F_{(1,59)}$=7.100, p=0.010, $\eta_p^2$=.107, $BF_{10}$ = 4.992; *Figure 1C and D*). This was also the case when considering only patients that had intact tactile perception on the contralesional forearm (i.e. could feel all the touch trials; N = 25, $F_{(1,43)}$=9.880, p=0.003, $\eta_p^2$ = .187, *Figure 1—figure supplement 1*; see Materials and methods section for details). A similar general tactile anhedonia (reduced pleasantness ratings) was observed in our patients as compared to the controls for imagined tactile pleasantness, when patients had to rate how pleasant it would be to be touched by pleasant and unpleasant fabric ($F_{(1,70)}$ = 22.348, p<0.001, $\eta_p^2$=.242, $BF_{10}$ = 550.118, *Figure 1—figure supplement 2*). However, no interaction between touch type and group was found (contralesional: $F_{(1,53)}$=0.393, p=0.533, $\eta_p^2$ = .007, $BF_{10}$ = 0.371, *Figure 1C*; ipsilesional: $F_{(1,59)}$=0.073, p=0.788, $\eta_p^2$ = .001, $BF_{10}$ = 0.287, *Figure 1D*; imagined tactile pleasantness: $F_{(1,70)}$=.061, p=0.806, $\eta_p^2$=.001, $BF_{10}$ = 0.270), suggesting that right hemisphere lesions in general do not necessarily lead to reduced CT pleasantness sensitivity, and confirming that any differential deficits in the pleasantness perception of CT-optimal versus CT-suboptimal touches at the individual level would relate to specific lesions rather than general stroke effects.

The present study aimed to investigate the lesion patterns and neuropsychological deficits that may be associated with the inability of certain stroke patients to distinguish the pleasantness of CT-optimal versus CT-suboptimal touches. Accordingly, CT pleasantness sensitivity was calculated as the difference between the pleasantness of CT-optimal and CT-suboptimal touches. As a convention, CT pleasantness sensitivity inferior or equal to zero is considered as low in CT pleasantness sensitivity (i.e. low CT affective touch perception; *Crucianelli et al., 2018*). Interestingly, none of the patients' demographic characteristics or, neuropsychological deficits correlated significantly with their CT pleasantness sensitivity, including education, anxiety and depression scores, as well as memory as measured by the MOCA memory subscale, and working memory as measured by the Digit Span (all p>0.1 and all $BF_{10}$ <1). Thus, low CT pleasantness sensitivity was not explained by other general cognitive and emotional deficits, as assessed in the present study. Moreover, there was no correlation between CT pleasantness sensitivities and tactile anhedonia on either forearm (as measure by the difference between the imagined pleasantness of pleasant and unpleasant material; $r_{31}$ = -.104, p=0.578, $BF_{10}$ = 0.259 for the contralesional forearm; $r_{36}$ = -.086, p=0.618, $BF_{10}$ = 0.234, for the ipsilesional forearm), nor with tactile acuity as measured by intensity ratings.

A VLSM analysis with CT pleasantness sensitivity on the contralesional forearm (differential pleasantness scores) as continuous predictor, controlling for lesions size, with a 0.01 FDR-corrected threshold, and considering only regions lesioned in at least 10 patients, revealed specific lesions in the rolandic operculum (see *Figure 2A*, *Figure 2—figure supplement 1A*, and *Table 1A*). Subcortically, the tracts of the superior corona radiata were involved. Importantly, running the same analysis

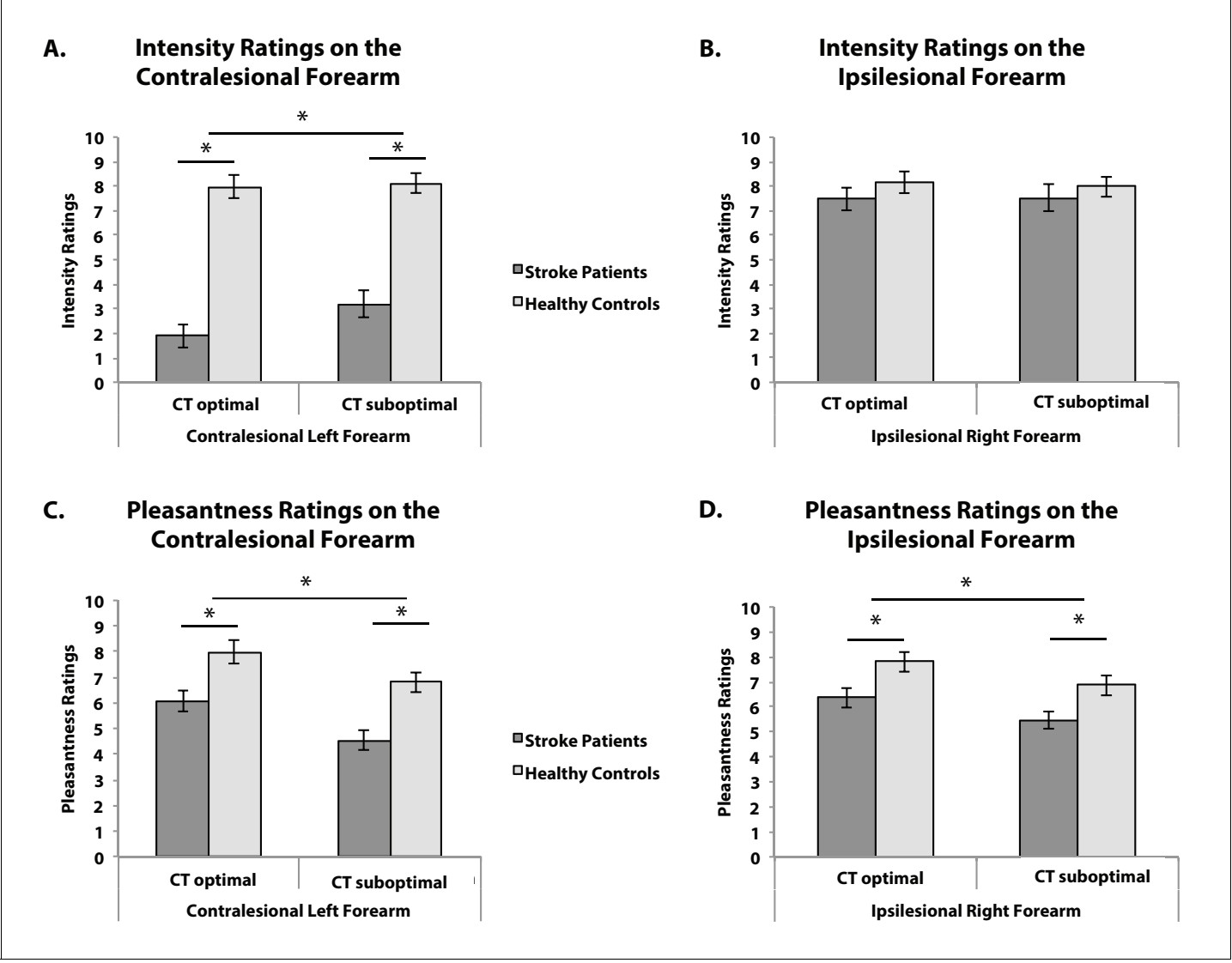

**Figure 1.** Behavioural Results. (**A**) Average intensity ratings on the contralesional left forearm ($N_{RH}$ = 39, $N_{HC}$ = 20), (**B**) Average intensity ratings on the ipsilesional right forearm ($N_{RH}$ = 20, $N_{HC}$ = 20), (**C**) Average pleasantness ratings on the contralesional left forearm ($N_{RH}$ = 35, $N_{HC}$ = 20), (**D**) Average pleasantness ratings on the ipsilesional right forearm ($N_{RH}$ = 41, $N_{HC}$ = 20), for CT-optimal and CT suboptimal touch. Stroke patients (RH) are depicted in dark gray, Healthy controls (HC) in light gray. Error bars represent the standard error of the mean. *depicts significant comparison, p<0.05.
The online version of this article includes the following figure supplement(s) for figure 1:

**Figure supplement 1.** Average pleasantness ratings on the contralesional left forearm for patients with intact tactile perception in dark gray ($N_{RH}$ = 25).
**Figure supplement 2.** Average pleasantness ratings for imaginary touch.

including only patients without sensory deficit on the left forearm (i.e. participants that rated all the trials as more intense than 2; N = 25) involves the same area but also extends to the posterior part of the insula (see *Figure 2B*, *Figure 2—figure supplement 1B* and *Table 1B*). This corroborates the importance of the posterior insula and the rolandic operculum in perceiving CT-optimal touch on the contralateral forearm as more pleasant than CT-suboptimal touch, particularly when other tactile pathways are intact.

In contrast, deficits in CT pleasantness sensitivity on the ipsilesional forearm were associated with lesioned voxels in the anterior part of the insula (including the adjacent regions, rolandic and frontal inferior operculum – see *Figure 2C*, *Figure 2—figure supplement 1C* and *Table 1C*). As patients' perception of the discriminatory, emotionally-neutral aspects of touch on the ipsilesional forearm was not affected (verified by the lack of difference in intensity ratings between healthy controls and

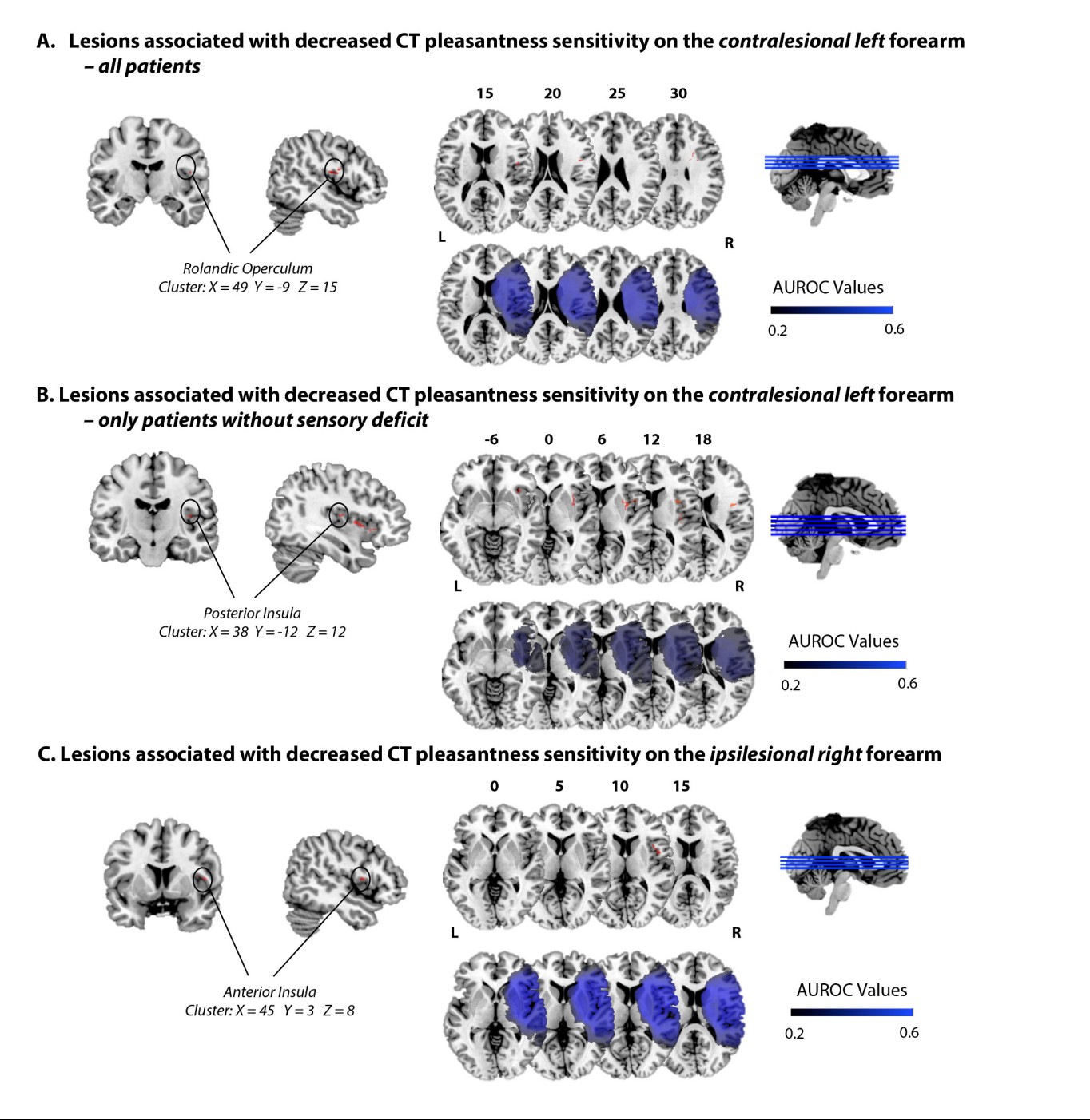

**Figure 2.** Lesions associated with decreased CT pleasantness sensitivity. (**A**) Lesions associated with decreased CT pleasantness sensitivity on the contralesional left forearm, in all patients (N = 35). (**B**) Lesions associated with decreased CT pleasantness sensitivity on the contralesional left forearm, only in patients without sensory deficit on the left (N = 25). (**C**) Lesions associated with decreased CT pleasantness sensitivity on the ipsilesional right forearm (N = 41). The numbers above the brain slices indicate the corresponding MNI axial coordinates. L = Left; R = Right; The second row represents heat maps of the voxels with power enough to detect significant results, at α = 0.01, FDR-corrected. Different colors represent the area under the ROC curve (AUROC) scores, ranging from 0.2 to 0.6.

The online version of this article includes the following figure supplement(s) for figure 2:

**Figure supplement 1.** Lesions Overlaps.

**Table 1.** Number of significant voxels (from the atlas of gray matter – AAL – and white matter – JHU – and NatBrainLab's atlas) resulting from the VLSM analyses.

A. with the CT pleasantness sensitivity scores for the *contralesional left* forearm as predictor, in all patients (N = 35); B. with the CT pleasantness sensitivity scores *for the contralesional left forearm* as predictor, only in patients without sensory deficit, N = 25; C. with the CT pleasantness sensitivity scores for the *ipsilesional right* forearm as predictor (N = 41).

**A. Lesions associated with decreased CT pleasantness sensitivity on the contralesional left forearm, in all patients (N = 35)**

| | Region | N$_{Voxels}$ | X | Y | Z | T-value |
|---|---|---|---|---|---|---|
| AAL | Unclassified | 104 | 43 | 1 | 19 | 2.88 |
| | Rolandic_Oper | 63 | 48 | -9 | 15 | 2.59 |
| JHU | Unclassified | 120 | 43 | 1 | 19 | 2.88 |
| | Superior_corona_radiata | 45 | 24 | 8 | 30 | 2.59 |
| NatBrainLab | Unclassified | 69 | 43 | 1 | 19 | 2.88 |
| | Arcuate_Anterior_Segment | 72 | 48 | -9 | 15 | 2.59 |
| | Corpus_Callosum | 11 | 22 | 7 | 28 | 2.56 |
| | Internal_Capsule | 15 | 25 | 5 | 27 | 2.56 |

**B. Lesions associated with decreased CT pleasantness sensitivity on the contralesional left forearm, only in patients without sensory deficit (N = 25)**

| | Region | N$_{Voxels}$ | X | Y | Z | T-value |
|---|---|---|---|---|---|---|
| AAL | Unclassified | 446 | 33 | 16 | -4 | 3.08 |
| | Frontal_Inf_Oper | 8 | 49 | 9 | 6 | 2.55 |
| | Frontal_Inf_Orb | 8 | 35 | 25 | -8 | 2.77 |
| | Rolandic_Oper | 88 | 37 | -4 | 20 | 2.57 |
| | Insula | 598 | 38 | −12 | 12 | 3.06 |
| | Putamen | 118 | 33 | -4 | 8 | 3.27 |
| | Heschl | 24 | 44 | −17 | 8 | 2.65 |
| JHU | Unclassified | 1254 | 33 | -4 | 8 | 3.27 |
| | Superior_corona_radiata | 8 | 26 | 8 | 24 | 2.57 |
| | External_capsule | 22 | 33 | -5 | 7 | 3.06 |
| | Superior_longitudinal_fasciculus | 6 | 32 | -6 | 24 | 2.57 |
| NatBrainLab | Unclassified | 1277 | 33 | -4 | 8 | 3.27 |
| | Arcuate_Anterior_Segment | 11 | 37 | -5 | 21 | 2.57 |
| | Inferior_Occipito_Frontal_Fasciculus | 1 | 37 | 2 | -8 | 2.54 |
| | Internal_Capsule | 1 | 26 | 8 | 24 | 2.57 |

**C. Lesions associated with decreased CT pleasantness sensitivity on the ipsilesional right forearm (N = 41)**

| | Region | N$_{Voxels}$ | X | Y | Z | T-value |
|---|---|---|---|---|---|---|
| AAL | Frontal_Inf_Oper | 59 | 42 | 9 | 9 | 2.76 |
| | Rolandic_Oper | 79 | 45 | 4 | 9 | 2.76 |
| | Insula | 32 | 45 | 3 | 8 | 2.70 |
| JHU | Unclassified | 170 | 45 | 4 | 9 | 2.76 |
| NatBrainLab | Unclassified | 170 | 45 | 4 | 9 | 2.76 |

patients, as well as patients' performance on a standardized somatosensory assessment; see Materials and methods, and *Figure 1B*), and as the left insula and somatosensory cortex of these patients were intact, these results suggest that the right anterior insula has a necessary role in the CT pleasantness sensitivity, even for the ipsilateral side of the body.

Additionally, as a control for a general pleasantness deficit, patients rated how pleasant it would be to be touched by a typically pleasant material and a typically unpleasant fabric. As done for CT pleasantness sensitivity, imagined tactile pleasantness sensitivity was computed as the difference

between pleasant and unpleasant materials pleasantness ratings, for each patient. We considered the same patients as for the CT pleasantness sensitivity VLSM analysis (N = 36 as we had missing data for 5 of them), and ran a VLSM analysis with this top-down tactile pleasantness sensitivity as predictor. This yielded significant voxels subcortically in the caudate, thalamus, putamen and pallidum, but crucially, not the insula, suggesting that the above results are specific to applied tactile stimuli and not more general pleasantness comparisons (see *Supplementary file 1*).

This lesion study aimed to investigate deficits in the perceived affectivity of CT-optimal touch. Our results suggest a causal role of the posterior contralateral opercular-insular cortex for the perception of CT-optimal touch as more pleasant than CT-suboptimal touch, offering support to previous, correlational, functional neuroimaging findings on the CT system (*Olausson et al., 2002*; *Morrison, 2016*). In addition, our findings reveal that the *right anterior* fronto-insular junction is necessary to perceive the pleasantness of CT-optimal touch as more pleasant than CT-suboptimal touch on the ipsilateral forearm. Thus, even when the left insula and somatosensory cortex are intact and hence presumably contralateral stimuli are processed in the left cortex (as also revealed by the intact detection of ipsilesional tactile stimuli in our patients), a right anterior insula lesion is enough to cause deficits in the perception of affective touch on the right forearm.

The present study has considered CT pleasantness sensitivity as the difference between the pleasantness of CT-optimal slow touch (3 cm/s) and CT-suboptimal fast touch (18 cm/s). Future studies should investigate whether the present findings replicate when using very slow touch instead of fast touch as CT-suboptimal touch, as very slow touch (<1 cm/s) also leads to suboptimal activation of the CT fibers (*Löken et al., 2009*). Moreover, the specificity of the present findings to CT fibers should be further investigated by comparing tactile stimulation on hairy (e.g. forearm) vs. glabrous skin (e.g. palm, that do not contain any CT fibers).

Taken together, our findings support previous findings about the functional organization and role of the human insula (*Craig, 2010*; *Cauda et al., 2011*; *Kurth et al., 2010*; *Heydrich and Blanke, 2013*; *Ronchi et al., 2015*; *Salomon et al., 2018*); see review by *Evrard (2019)*, on recent findings on the organization of the insula in non-human primates), consisting of specialized substrates organized in a posterior to anterior structural progression, with posterior parts representing the primary cortical representations of interoceptive stimuli from contralateral body parts and more anterior parts, tested here in the right hemisphere, acting as integration areas for sensory signals and contextual cues ultimately leading to interoception. Indeed, present findings are consistent with the growing evidence considering CT-afferents as sharing more characteristics with interoceptive (i.e. related to the sense of the physiological condition of one's own body; *Ceunen et al., 2016*), rather than exteroceptive, modalities (*Björnsdotter et al., 2010*), in light of their contribution to the maintenance of our sense of self (*Crucianelli et al., 2018*). Moreover, our findings address existing debates about hemispheric laterality and interoception, with a right-hemisphere dominance in interoceptive integration of both contra- and ipsilateral signals (*Kann et al., 2016*; *Khalsa et al., 2009*; *Salomon et al., 2016*; *Garfinkel and Critchley, 2013*), although the VLSM method has known intrinsic limitations, and we cannot exclude the possible role of the left insula in affective touch perception, nor the impact of lesions of the right hemisphere in disconnecting tracts towards the left hemisphere. Furthermore, as VLSM methods preclude direct comparison between CT pleasantness sensitivity deficits on the contralesional and ipsilesional forearm at the brain level, future studies should investigate further the posterior-anterior insula segregation in relation to affective touch as an interoceptive modality.

## Materials and methods

### Subjects and clinical investigation

Fifty-nine, unilateral, right-hemisphere-lesioned stroke patients (mean age: 65.86 ± 14.12 years; age range: 38–88 years; 31 females) were recruited from consecutive admissions to seven stroke wards as part of a larger study using the following inclusion criteria: (i) imaging-confirmed first ever right hemisphere lesion; (ii) contralateral hemiplegia; (iii) < 4 months from symptom onset; (iv) no previous history of neurological or psychiatric illness; (v) > 7 years of education; (vi) no medication with significant cognitive or mood side-effects (e.g. pregabalin, lamotrigine); (vii) no language impairments that precluded completion of the study assessments; and (viii) right handed. All participants gave written,

informed consent to take part in the study. The local National Health System Ethics Committees approved the study, which was carried out in accordance to the Declaration of Helsinki.

All patients underwent a thorough neurological and neuropsychological examination. Premorbid intelligence was assessed using the Wechsler Test of Adult reading (WTAR; *Wechsler, 2001*). Post-morbid, general cognitive functioning, including long-term verbal recall was assessed using the Montreal Cognitive Assessment (MoCA; *Nasreddine et al., 2005*). The Medical Research Council scale (MRC; *Saunders, 1986*) was used to assess limb motor strength. Proprioception was assessed with eyes closed by applying small, vertical, controlled movements to three joints (middle finger, wrist and elbow), at four time intervals (correct = 1; incorrect = 0; *Vocat et al., 2010*). Working memory was assessed using the digit span task from the Wechsler Adult Intelligence Scale III (*Wechsler, 1997*). The Hospital Depression and Anxiety Scale (HADS; *Zigmond and Snaith, 1983*) was used to assess anxiety and depression. Executive and reasoning abilities were assessed using the Frontal Assessment Battery (FAB; *Dubois et al., 2000*). Four subtests of the Behavioural Inattention Test (BIT; *Wilson et al., 1987*) were used to assess visuospatial neglect. Personal neglect was assessed using the 'one item test' (*Bisiach et al., 1986*) and the 'comb/razor' test (*McIntosh et al., 2000*).

Twenty age-matched healthy control subjects were recruited and tested with the same behavioural paradigm in order to assess the specificity of deficits in the patient group (healthy control group; 63.05 ± 12.12 years; age range: 46–87 years; 11 females). Patients' demographic characteristics and their performance on standardized neuropsychological tests and how they compared to the healthy sample are summarized in *Table 2*.

## Design and Predictions

The present study aimed to investigate the neuroanatomical bases of affective touch. To this aim, we compared a large cohort of right hemisphere stroke patients to healthy controls, and explored how deficits in affective touch perception are linked with specific brain lesions. We applied an affective touch paradigm that manipulated three factors: i) the velocity of the touch applied (slow, CT-optimal, affective touch vs. fast, CT-suboptimal, neutral touch); ii) the forearm the touch was applied to (right, ipsilesional vs. left, contralesional); iii) and the group of participant (Stroke patients vs. Healthy controls). For each type of touch we recorded two measures: 1) intensity ratings and 2) pleasantness ratings. We also asked participants to rate the pleasantness of imagined touch with either a smooth material (velvet) versus a rough material (sandpaper), to control for top-down effects; and general tactile anhedonia due to right hemisphere stroke.

To investigate the neuroanatomical bases of affective touch, we conducted two main voxel-based, lesion-symptom mapping analyses, separately for each forearm, using as predictors the CT pleasantness sensitivity (difference between average pleasantness ratings for CT-optimal touch and CT-suboptimal touch). In addition to the main analyses we also ran a control analysis, using the difference between imagined pleasantness ratings of pleasant (velvet) and unpleasant (sandpaper) material as predictors, to control for potential top-down affective deficit. Finally, a lesion overlap was calculated to create a color-coded overlay map of lesioned voxels across participants with negative or null CT pleasantness sensitivity on each forearm.

Given our patients' lesions to several perisylvian regions of the right hemisphere previously associated with somatosensory perception (*Dijkerman and de Haan, 2007*; *Preusser et al., 2015*), we expected that our patients would have, on average, reduced ratings of both touch intensity and pleasantness in comparison to healthy controls, and specifically in the contralesional left forearm. However, we did not expect a general right stroke effect on pleasantness sensitivity to CT affective touch (defined as the pleasantness difference between CT-optimal and CT-suboptimal velocities), given the assumed neurophysiological specificity of the CT system. Instead, we expected that lesions involving mainly the right posterior insula (*Morrison, 2016*) would lead to a lack of CT pleasantness sensitivity, particularly on the contralesional forearm. Moreover, as some authors have proposed that the right hemisphere, and particularly the right anterior insula, has a crucial role in interoceptive awareness for the entire body (*Craig, 2009*; *Critchley et al., 2004*; *Kann et al., 2016*; *Khalsa et al., 2009*; *Salomon et al., 2016*), we expected also to find some causal role of ipsilateral areas (right hemisphere regions after touch on the right forearm) and particularly the right anterior insula in the perception of affective touch on the ipsilesional forearm.

**Table 2.** Summary of demographics and neuropsychological data.

Description: Nottingham = Light Touch subscale of the Revised Nottingham Sensory Assessment (rNSA; *Lincoln et al., 1998*; score overall for each arm with 0: no sensation; 1: slightly impaired; 2: no deficit); MRC = Medical Research Council scale (*Saunders, 1986*); MOCA = The Montreal Cognitive Assessment (*Nasreddine et al., 2005*); FAB = Frontal Assessment Battery (*Dubois et al., 2000*); Pre-morbid IQ-WTAR = Wechsler Test of Adult Reading (*Wechsler, 2001*); HADS = Hospital Anxiety and Depression scale (*Zigmond and Snaith, 1983*); Comb/razor test = tests of personal neglect (*McIntosh et al., 2000*); Bisiach one item test = test of personal neglect; line crossing, star cancellation, copy and representational drawing = conventional sub-tests of Behavioural Inattention Test (*Wilson et al., 1987*). Dashed line indicates not applicable. Due to several clinical constraints (e.g. fatigue, acceptance and time constraints), we have a number of missing data on these tests. Specific numbers are indicated in the right column. $N_{RH}$ = number of right hemisphere stroke patients having fully completed the corresponding test. $N_{HC}$ = number of healthy controls having fully completed the corresponding test. [*] Significant difference between groups, p<0.05.

| | Stroke Patients – RH (N = 59; 31 females) | | Healthy Controls - HC (N = 20, 11 females) | | Mann-Whitney Test | $N_{RH}/N_{HC}$ |
|---|---|---|---|---|---|---|
| | **Mean** | *SD* | **Mean** | *SD* | | |
| Age (years) | 65.86 | 13.87 | 63.05 | 12.12 | *U(78)=514.00, Z = -.857, p=0.391* | *N = 59/20* |
| Education (years) | 11.40 | 2.87 | 14.75 | 2.82 | *U(70)=211.50, Z = −3.906, p<0.001[*]* | *N = 52/20* |
| Days from onset | 16.95 | 18.68 | - | - | | |
| Orientation | 2.80 | 0.41 | - | - | | |
| Nottingham on left arm (max 2) | 0.66 | 0.78 | - | - | | |
| Nottingham on right arm (max 2) | 2 | 0 | - | - | | |
| Proprioception (max 9) | 5.10 | 2.64 | - | - | | |
| MRC Left upper limb | 0.30 | 0.75 | - | - | | |
| Digit span forwards | 5.95 | 1.40 | 6.58 | 1.83 | *U(66)=279.50, Z = 0.936, p=0.349* | *N = 56/12* |
| Digit span backwards | 3.50 | 1.55 | 4.75 | 1.28 | *U(66)=177.00, Z = −2.621, p=0.009[*]* | *N = 56/12* |
| MOCA | 19.85 | 5.18 | 28.19 | 1.92 | *U(45)=5.50, Z = −4.271, p<0.001[*]* | *N = 39/8* |
| MOCA memory subscale | 2.92 | 1.78 | 4.00 | 1.60 | *U(45)=95.00, Z = −1.769, p=0.077* | *N = 39/8* |
| Premorbid IQ-WTAR | 34.00 | 9.35 | 49.11 | 1.69 | *U(25)=3.00, Z = −4.037, p<0.001[*]* | *N = 18/9* |
| HADS depression | 5.75 | 3.49 | 3.13 | 2.19 | *U(50)=150.00, Z = −2.593, p=0.010[*]* | *N = 37/18* |
| HADS anxiety | 8.02 | 4.33 | 6.06 | 3.01 | *U(50)=208.00, Z = −1.409, p=0.159* | *N = 37/18* |
| FAB total score | 11.38 | 4.02 | - | - | | |
| Comb/razor test bias (%bias) | −23.37 | 27.06 | - | - | | |
| Bisiach one item test | 0.47 | 0.68 | - | - | | |
| Line crossing (max 36) | 22.56 | 11.85 | - | - | | |
| Star cancelation (max 54) | 29.93 | 18.23 | - | - | | |
| Copy | 0.87 | 1.20 | - | - | | |
| Representational drawing | 0.62 | 0.93 | - | - | | |
| Line bisection | 2.87 | 3.05 | - | - | | |

### Affective touch protocol

Tactile stimulation followed a previously validated protocol (*Crucianelli et al., 2013*; *Crucianelli et al., 2018*; *Gentsch et al., 2015*; *Mohr et al., 2017*; *Kirsch et al., 2018*), including both 'imagined' and actual touch questions. Specifically, first a 9 cm x 4 cm area of skin stimulation was marked on both forearms and patients were familiarized with the vertical rating scales (to minimize the effects of neglect; we also always ensured the participants could see the scale and read it aloud to facilitate them), anchored at '0 - not at all' and '10 - extremely'. We first sampled top-down, prior beliefs about tactile pleasantness by asking two hypothetical questions about imagined touch: 'How pleasant would it be to be touched by velvet on your skin' (typically pleasant) and 'How pleasant would it be to be touched by sandpaper on your skin?' (typically unpleasant). Participants

were asked to answer using the vertical 0 to 10 pleasantness scale. No other instruction was given to the participants (neither speed nor pressure of the imagined touch).

We then explained that actual tactile stimuli would be delivered on the marked forearm areas, while participants were blindfolded, and instructed to remain still and to focus on both the intensity and pleasantness of the touch they were experiencing (*Figure 3*). Tactile stimuli were administrated by a 4 cm wide soft make up brush made from natural hair (Natural hair Blush Brush, No. 7, The Boots Company). Brush strokes were administered by a trained female experimenter in proximal-to-distal direction with the brush held in a perpendicular position, with the edges of the brush tracking the width of the testing area to control for pressure. Every touch condition lasted for 3 s; with an inter-stimuli interval of at least 30 s. After each touch, participants were asked to answer two questions: first 'How well did you feel the touch?' (i.e. touch intensity rating), and if they felt the touch (i.e. reporting an intensity rating >0), they were asked 'How pleasant was the touch?' (i.e. touch

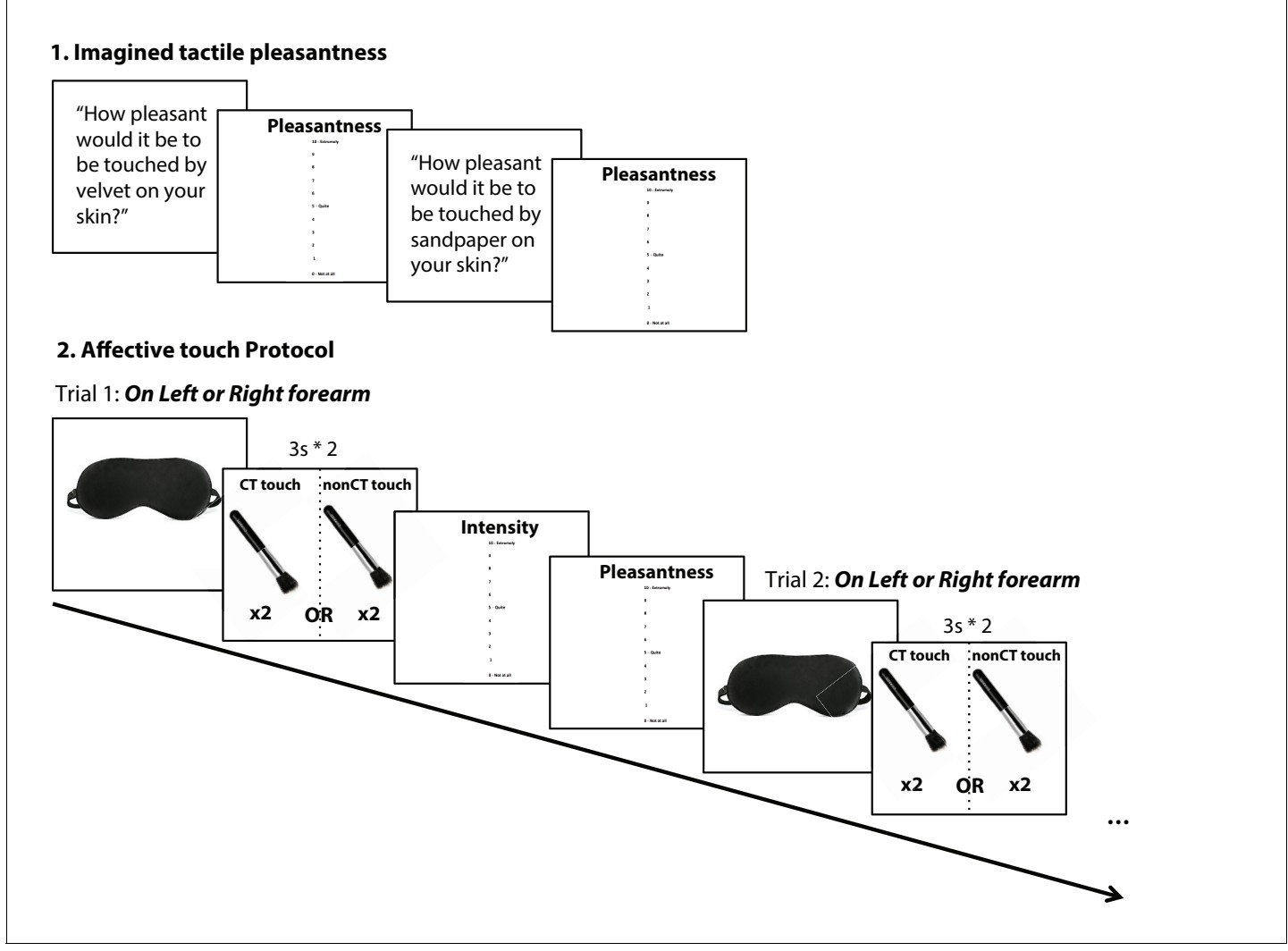

**Figure 3.** Experimental design and timeline. 1. Participants were first asked to answer two hypothetical questions about imagined touch: 'How pleasant would it be to be touched by velvet on your skin' (typically pleasant) and 'How pleasant would it be to be touched by sandpaper on your skin?' (typically unpleasant). Participants were asked to answer using the vertical 0 to 10 pleasantness scale. 2. Participants were then asked to put on a blindfold at the onset of each trial before the experimenter delivered the touch on the left or right forearm at CT-optimal (CT touch) or CT-suboptimal velocities (nonCT touch; pseudorandomized), each touch lasted for 3 s and was repeated twice with a one second break in between. After each touch, blindfold was removed so participants could rate the touch on two scales: Intensity = How well they felt the touch; and Pleasantness = How pleasant was the touch, each on a vertical scale ranging from 0, not at all, to 10, extremely. After ratings were recorded, the participant was asked to put the blindfold back before starting the next trial.

pleasantness rating), using the above described 0 to 10 vertical scale. Tactile stimuli were delivered at two different velocities on the participant's left and right forearm: CT-optimal speed (3 cm/s, known to activate CT fibers optimally; one stroke over the 9 cm long area) and CT-suboptimal speed (18 cm/s, known to activate CT fibers to a lesser degree, suboptimally; *Gentsch et al., 2015*; six strokes). Each condition was repeated 6 times, leading to a total of 24 trials – delivered in a pseudor-andomized order. The experiment was split into three blocks to avoid fatigue; short breaks were taken after a set of 8 trials (2 repetitions of each condition in each block).

All patients had intact sensation on the right ipsilesional forearm (i.e. rated the intensity of tactile stimuli as greater than zero in all the trials, irrespective of velocity, and had intact sensation on this side according to a standardized assessment; the Revised Nottingham Sensory Assessment [rNSA; *Lincoln et al., 1998*]) but as predicted, on the contralesional side, some patients (40.7%, N = 24) were not able to perceive the tactile stimuli (corroborated also by the above standardized somato-sensory assessment), and therefore gave a rating of zero on the intensity scale, and were not asked to provide pleasantness ratings. Thus, pleasantness ratings were available only from the remaining 35 patients who were able to perceive the intensity of most contralesional tactile stimuli in our paradigm.

## Behavioural data analysis

We investigated the effect of right hemisphere lesions on the perception of touch intensity and pleasantness, on the contralesional and ipsilesional forearm separately, by comparing stroke patients and healthy controls intensity and pleasantness ratings in turn. As the data were normally distrib-uted, separate ANOVAs were run with touch type (CT-optimal vs. CT-suboptimal) and group (stroke patient vs. healthy controls) as factors, for each rating type and each forearm. An additional ANOVA comparing stroke patients and healthy controls was conducted for the imagined tactile pleasantness ratings (velvet vs. sandpaper).

We were able to collect contralesional touch intensity ratings on only 39 out of the total sample of 59 patients due to an administrative error (the experimenter took binary, 'yes' or 'no' responses to the tactile stimuli instead of using the rating scale in the remaining patients). For the same reason, we only had ipsilesional touch intensity ratings for CT-optimal touch on 36 and CT-suboptimal touch on 20 patients. This unfortunately meant that our sample was reduced to 39 patients for the analyses of intensity ratings on the contralesional forearm and of 20 patients for the ipsilesional forearm.

We were able to record pleasantness ratings for contralesional forearm touch on 35 and 39 patients for CT-optimal and CT-suboptimal touch velocities respectively (data of 21 and 13 patients were missing due to the fact that patients did not feel the touch and gave an intensity rating of 0; the remaining 3 and 8 missing data were due to an administrative error). For the right ipsilesional forearm, pleasantness ratings of 56 and 41 patients were recorded at CT-optimal and CT-suboptimal touch velocities respectively. Thus, the sample of the analysis of touch pleasantness was of 35 patients for the contralesional forearm and of 41 patients for the ipsilesional forearm.

Moreover, as supplementary analyses, we also considered patients with intact tactile perception on the contralesional forearm. For these analyses, only patients that gave intensity ratings above two were included (N = 25).

We used both frequentist and Bayesian statistics to assess the observed effects, depending on the aim and hypothesis in each case. The complementary use of these two statistical approaches is recommended by a number of authors to facilitate a fuller understanding of the data (see e. g. *Dienes and Mclatchie, 2018*; *Dienes, 2014*; *Jarosz and Wiley, 2014*; *Howard et al., 2000*). Bayesian statistics were performed in order to allow further interpretation of the observed effects, in particular, the extent to which data provided support for the alternative versus null hypotheses. Bayes Factors (BF10) provide a continuous measure that indicates the relative strength for the null versus alternative hypotheses (i.e. the number of times more likely the data are under the alternative than the null hypothesis), and were used as a means of interpreting evidence for each hypothesis, using benchmarks provided by *Jeffreys (1939)*. We interpreted a BF10 >3 as substantial evidence for the alternative hypothesis, a BF10 <0.3 as substantial evidence in favour of the null hypothesis, and 0.3 < BF10<1 as anecdotal evidence in favour of the null hypothesis (see *Dienes, 2014*). Bayes Factor were computed using JASP (*JASP Team, 2019*). JASP (Version 0.10).

## Lesion mapping methods and analyses

Routinely acquired clinical scans obtained on admission (<2 days post stroke) were collected for the 59 patients (49 via computerized tomography, CT; and 10 via magnetic resonance imaging, MRI). We note that testing patients in the acute post-stroke phase entails challenges but avoids any confounds relating to plasticity and functional reorganization (*Baier et al., 2014*; *de Haan and Karnath, 2018*). The patient's lesion was mapped by means of the MRIcron software (*Rorden and Brett, 2000*) on the standard T1-weighted MRI template (ICBM152) of the Montreal Neurological Institute (MNI) coordinate system. Lesions from these scans were segmented and co-registered using a manual procedure, as this method remains the best methods to date for lesion mapping of clinical scans and shown to be more accurate than automatized methods (*Maier et al., 2015*; *de Haan and Karnath, 2018*; *Liew et al., 2018*). Two expert clinicians, blind to the hypotheses of the study, outlined the lesions. In the case of disagreement of two lesion plots, the opinion of a third, expert anatomist was requested. Scans were registered to the T1-weighted MRI scan template (ICBM152) of the Montreal Neurological Institute, furnished with the MRIcron software (ch2, http://www. cabiatl.com/mricro/mricron/index.html). First, the standard template (size: 181 × 217×181 mm, voxel resolution: 1 mm$^2$) was rotated on the three planes in order to match the orientation of the patient's MRI or CT scan. Lesions were outlined on the axial slices of the rotated template. The resulting lesion volumes were then rotated back into the canonical orientation, in order to align the lesion volumes of each patient to the same stereotaxic space. Finally, in order to exclude voxels of lesions outside white and gray matter brain tissue, lesion volumes were filtered by means of custom masks based on the ICBM152 template.

The statistical contribution of lesion location to CT pleasantness sensitivity and imagined tactile pleasantness deficits was tested using voxel-based lesion symptom mapping (VLSM), using the behavioral scores as continuous predictor. The statistical process performed in voxel-based lesion–symptom mapping (*Bates et al., 2003*) consists of the following steps: at each voxel of the spatially standardized scan images, patients are divided into two groups according to whether they did or did not have a lesion affecting that voxel. Behavioral scores are then compared for these two groups with a t-test, yielding a single-tailed p-value for each voxel. Normal t-tests were used as the behavioural data entered in the VLSM models were normally distributed (*de Haan and Karnath, 2018*). This method allows controlling for lesion size, which is included as a covariate of non-interest. Note that to avoid spurious results due to low numbers of lesioned voxels, only voxels lesioned in at least 10 participants were tested. This results in color-coded VLSM maps that represent voxels where patients with lesions show a significantly different behavioral score from those whose lesions spared that voxel at an α level of 0.01 after correction for multiple comparisons using the false discovery rate (*Curran-Everett, 2000*). Software to perform VLSM (*Bates et al., 2003*; https://aphasialab.org/vlsm/) was run using MATLAB R2016b (Mathworks, Inc). It is to note, that in accordance with recent recommendation by *de Haan and Karnath (2018)*, as no correlations were found between CT pleasantness sensitivity (or pleasantness ratings) and any of the neuropsychological scores that differed between healthy controls and right hemisphere stroke patients (HADS Depression scale, Digit Span backward, MOCA memory scale, and Premorbid IQ-WTAR), none of these variables could be considered as nuisance variables and were not considered in the VLSM lesion analyses.

Each analysis was conducted separately for the contra- and the ipsilesional forearm, and only regions of more than 10 voxels that passed the set 0.01 FDR-corrected threshold were considered in the discussion. VLSM results were visualized in MRIcron. Three anatomical templates served to identify gray and white matter region labels: the 'automated anatomical labelling' (AAL) template (*Tzourio-Mazoyer et al., 2002*), the JHU white-matter tractography atlas, (*Mori et al., 2005*), and the 'NatBrainLab' template of the 'tractography based Atlas of human brain connections Projection Network' (Natbrainlab, Neuroanatomy and Tractography Laboratory; *Catani and de Schotten, 2012*; *Thiebaut de Schotten et al., 2011*).

## Acknowledgements

We thank the all the stroke patients for their kindness and willingness to take part in the study, as well as the healthy participants. We are also particularly grateful to Sonia Ponzo, Amanda Hornsby, and Arturo Kerbel, for their help with patient recruitment and testing.

## Additional information

### Funding

| Funder | Grant reference number | Author |
|---|---|---|
| European Research Council | ERC-2012-STG GA313755 | Aikaterini Fotopoulou |
| Ministry of Education, University and Research | PRIN 20159CZFJK | Valentina Moro |
| University of Verona | Bando di Ateneo per la Ricerca di Base 2015 project MOTOS | Valentina Moro |

The funders had no role in study design, data collection and interpretation, or the decision to submit the work for publication.

### Author contributions

Louise P Kirsch, Conceptualization, Formal analysis, Investigation, Visualization, Methodology, Project administration; Sahba Besharati, Christina Papadaki, Laura Crucianelli, Conceptualization, Investigation, Methodology; Sara Bertagnoli, Investigation, Drawing of patients' lesion; Nick Ward, Helped recruiting the stroke patients on his ward; Valentina Moro, Investigation, Drawing of patients' lesions; Paul M Jenkinson, Conceptualization; Aikaterini Fotopoulou, Conceptualization, Supervision, Funding acquisition, Methodology

### Author ORCIDs

Louise P Kirsch (iD) https://orcid.org/0000-0002-8418-776X
Aikaterini Fotopoulou (iD) https://orcid.org/0000-0003-0904-7967

### Ethics

Human subjects: All participants gave written, informed consent to take part in the study and to publish. The local National Health System Ethics Committees approved the study (REC:05/Q0706/218), which was carried out in accordance to the Declaration of Helsinki.

### Decision letter and Author response

Decision letter https://doi.org/10.7554/eLife.47895.sa1
Author response https://doi.org/10.7554/eLife.47895.sa2

## Additional files

### Supplementary files

• Supplementary file 1. Number of significant voxels (from the atlas of gray matter – AAL – and white matter – JHU – and NatBrainLab's atlas) resulting from the VLSM analysis with the general pleasantness sensitivity scores (velvet-sandpaper average pleasantness ratings), N = 36. As control for a general pleasantness deficit, patients rated how pleasant it would be to be touched by a typically pleasant material (i.e. velvet, $M_{pleasantness\ rating}$ = 6.91, SD = 1.88) and a typically unpleasant fabric (i.e. sandpaper, $M_{pleasantness\ rating}$ = 0.33, SD = 0.93). Similarly, as for CT pleasantness sensitivity, top-down tactile pleasantness sensitivity was computed as the difference between pleasant (velvet) and unpleasant pleasantness ratings (sandpaper), for each patient. We considered the same patients as for the CT pleasantness sensitivity VLSM analysis (N = 36 as we had missing data for 5 of them) and ran a VLSM analysis with this top-down tactile pleasantness sensitivity.

• Transparent reporting form

### Data availability

The data that support the findings of this study are available on the Open Science Framework (https://osf.io/fyrwc/).

The following dataset was generated:

| Author(s) | Year | Dataset title | Dataset URL | Database and Identifier |
|---|---|---|---|---|
| Kirsch LP, Besharati S, Papadaki C, Crucianelli L, Bertagnoli S, Ward N, Moro V, Jenkinson PM, Fotopoulou A | 2019 | Affective Touch Lesion Study | https://osf.io/fyrwc/ | Open Science Framework, fyrwc |

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
