## [Decision Letter]

**Acceptance summary:**

The authors map brain circuits involved in the perception of affective touch, the tactile sensation associated with affiliative social contact. The authors identified a large cohort of patients with brain lesions in the insular cortex, and used clinical behavioral assays across patients to map regions involved in affective touch sensation. These studies advance our understanding of neural circuits that process sensations of positive valence in the somatosensory system.

**Decision letter after peer review:**

Thank you for submitting your article "Damage to the right insula disrupts the perception of affective touch" for consideration by *eLife*. Your article has been reviewed by Christian Büchel as the Senior Editor, a Reviewing Editor, and two reviewers. The reviewers have opted to remain anonymous.

The reviewers have discussed the reviews with one another and the Reviewing Editor has drafted this decision to help you prepare a revised submission.

Summary:

This manuscript examines whether sensations of pleasant touch are altered in stroke patients with lesions in the insular cortex. A large patient cohort was analyzed, and a correlation was observed, with differences observed between anterior and posterior cortical regions.

Essential revisions:

*Reviewer #1:*

The paper shows that stroke patients with insular lesions rate the pleasantness of arm-stroking less than patients with lesions elsewhere. Arm stroking was conducted in a manner known to optimally stimulate firing of unmyelinated 'C-tactile' fibres, a channel of cutaneous somatosensation most associated with affective experience, including pleasant stroking, and less pleasant tickle and itch. Previous studies (e.g. Olausson et al., 2002) have shown the preservation of C-tactile fibres in individuals with peripheral cutaneous somatosensory (demyelinating) neuropathy can lead to patients with limb anesthesia to experience cutaneous sensation, an effect related to insular rather than primary somatosensory cortical activation. This paper adds to this knowledge but nevertheless can be improved.

1) The nomenclature could be tighter: The 3-10cm/s stroking stimulation is called CT optimal or suboptimal, based on electrophysiological recording (can this evidence be cited). However, calling reported pleasantness 'CT pleasantness sensitivity' is stretching this link as there is no direct evidence that people are reporting an exclusive readout of CT fibres. Better to refer to touch (or stroking) pleasantness.

2) The papers cited relating pleasantness to optimal CT stimulation (e.g. Gentsch, 2015) were undertaken in non-clinical individuals. Patients may not show the same profile of stroking rate / pleasantness judgment, e.g. may feel pleasantness at a different rate of stroking but it is assumed that the normative relationship holds.

3) It is not actually that clear what the higher-level predictions (hence conclusions) actually were. Some experiences of touch are more neutral than other experiences of touch; while somatosensory cortex is not associated with construction / appraisal of emotional meaning, parts of insula and other regions are implicated in affective judgements, in part through association with interoceptive processes (e.g. bradycardia associated with affiliative interaction). Is the question then whether there is selective deficit in the experience of positive affect related to speed of touch in a brain-damaged patient group that can be localised to areas implicated in affect? There are multiple aspects to this question, so it ought to be clearly motivated.

4) Continuing the point above, does the investigation of right hemisphere regions associated with the perceived affective impact of CT-optimal touch condense to whether affective experience is compromised by focal (insular) lesions; whether stroking sensation, rather than not discriminative touch is compromised by focal lesions, or whether the affective aspects of stroking sensation can uncoupled from non-affective intensity of such sensation. This is relevant as, for example, there are no equivalent (exteroceptive or interoceptive) control for pleasant sensation, so general or specific inferences are constrained. Affective reactivity to other external stimuli may also be an important individual difference, although medication with significant cognitive or mood side effect were excluded (it would be useful to state explicitly that this included β blockers, pregabalin, lamotrigine). Nevertheless, mood was recording and it is unclear how tactile anhedonia related to negative affectivity in the patients.

5) The hypothetical touch rating (velvet vs sandpaper) protocol could be mentioned in the Introduction as it pops out unexpectedly in the Results section. The rationale should be more explicit. (Is it a sub hypothesis about perceptive loss leading to imaginative loss for this channel of sensation?). In the Materials and methods section, it could be usefully described in more detail: were people asked to imagine velvet and sandpaper drawn across the skin at different velocities in order to mirror the effects of CT optimal / suboptimal stimulation. Would an interaction be expected?

6) There is mention of interoception and interoceptive awareness e.g. in the Discussion section. The consensus definition of interoception is the sensation of the internal state of the body. If the authors maintain Craig's proposal that stroking and tickle sensation are honorary interoceptive sensations by virtue of having un- or poorly- myelinated fibres and sharing in part the same spinal and perhaps central pathways as viscerosensory afferents then this should be strongly and explicitly argued. If this is argued that the core aspect is that these sensations are affective/emotive then the issue about prediction / hypothesis underlying the experimental design is relevant. The term interceptive awareness has evolved since Garfinkel, 2013 in the light of consciousness science equating awareness with metacognitive insight, interoceptive sensibility being one term used for reported perception or misperception of visceral sensation.

7) Anatomical understanding of insula has been enhanced greatly through the work of Evrard. This could be reasonably cited.

*Reviewer #2:*

The manuscript describes a lesion study in 59 patients investigating the impact of insular damage on CT functioning. The study is very interesting, very timely and of potential high contribution for the field. The methods are sound. The Introduction, Discussion section and Results section are quite short. This could be a benefit, but it makes the paper hard to understand at several places.

My main concern is the mismatch between hypothesis, result and discussion. The authors hypothesize the pleasantness difference (CT optimal- suboptimal velocities) to be higher when patients are contralateral stroked than in other conditions (healthy controls or patients ipsilateral), This would be a 3-way interaction (velocity by group by side) which is not tested. In addition, Figure 1 does not support this hypothesis. Anyhow, the authors write in the Discussion section, that the hypothesis is supported.

Support for the hypothesis comes from the VLSM approach. I needed to reread the results several times in order to understand this. Especially as the methods are described after the results, it is unclear what the authors actually did. It would be much easier, if the authors would introduce the approach and the corresponding hypothesis in the Introduction. Results and Discussion section: what kind of patients are the ones with intact sensation? How did you measure this? In the Results and Discussion section, the authors introduce a comparison between hypothetical velvet stroking and sandpaper stroking. This is very surprising and could also be introduced better in the introduction part. I would actually remove this part, which is in my eyes unnecessary for the hypothesis but confusing for the reader.

The generally reduced pleasantness scores in lesion patients are most likely due to the enhanced depression and reduced attention in this group, a confounder analysis would clarify this.

[Editors' note: further revisions were suggested prior to acceptance, as described below.]

Thank you for submitting your article "Damage to the right insula disrupts the perception of affective touch" for consideration by *eLife*. Your article has been reviewed by Christian Büchel as the Senior Editor, a Reviewing Editor, and two reviewers. The following individuals involved in review of your submission have agreed to reveal their identity: Hugo Critchley (Reviewer #1).

The reviewers have discussed the reviews with one another and the Reviewing Editor has drafted this decision to help you prepare a revised submission.

The reviewers noted a substantial improvement in the revised manuscript, but a few issues persisted that should be addressed. Please see the reviewer comments below.

*Reviewer #1:*

This is an interesting and novel study of regional brain lesions and aspects of touch perception preferentially associated with unmyelinated peripheral cutaneous sensory nerve transmission.

I had previously raised a number of points, which the authors have taken time and consideration to address in full.

I suggest no further changes and anticipate the paper will attract interest with impact from *eLife* readership.

*Reviewer #2:*

The revised manuscript improved in terms of understandability and the authors clearly put effort in the rebuttal. However, my concerns are not eliminated.

I still have some questions about the "pleasantness sensitivity"

- It is not clear to me why the authors used 18cm/s as a control for suboptimal CT activation. The microneurography studies are far from clear for this velocity and 30cm/s would be the standard with supporting microneurography data.

- How did the authors deal with those cases where roof or bottom effects were observed (e.g. rating all touch highly positive or negative)?

- How many of the healthy controls have a "negative pleasantness sensitivity"? And how many do not seem to differentiate between 3 and 18cm/s (no difference or very minor difference in pleasantness ratings)?

I may have missed it, but I cannot find the place where the authors show that the construct they call CT pleasantness sensitivity differs between patients with posterior insula lesions and healthy controls. Instead, the authors report differences between patients with various lesions. This does not relate to the abstract or to the hypothesis.

I don't understand why the authors did not control for potential sources of variance in the VTLM. The authors argue that "However, as explained in point 4 above, no correlations were found between CT pleasantness sensitivity (or pleasantness ratings) and any of the neuropsychological tests that differed between healthy controls and right hemisphere stroke patients (HADS Depression scale, Digit Span backward, MOCA and Premorbid IQ_WTAR). " However, the main observation the authors report is not between healthy controls and patients, but within groups of patients.

---

## [Author Response]

Essential revisions:Reviewer #1:1) The nomenclature could be tighter: The 3-10cm/s stroking stimulation is called CT optimal or suboptimal, based on electrophysiological recording (can this evidence be cited). However, calling reported pleasantness 'CT pleasantness sensitivity' is stretching this link as there is no direct evidence that people are reporting an exclusive readout of CT fibres. Better to refer to touch (or stroking) pleasantness.

We thank the reviewer for the positive assessment of our study’s scope and their suggestions for improving the manuscript.

We had omitted citations to microneurography studies due to the limit on citations the journal requires as well as word restrictions for Short Reports, but following the reviewers’ suggestion we now cite the following studies: Vallbo and Hagbarth, 1968; Vallbo et al., 1993, 1999; Nordin, 1990 (Introduction).

We agree that there is no evidence or current technical way to provide a tactile stimulus to individuals that activates only the CT fibers, and we now explicitly acknowledge this in the manuscript (Introduction and below). However, in this study we have used a psychophysical procedure, with control procedures and measures for top-down effects (see manuscript and below) and a related differential measure (the felt pleasantness difference between tactile stimuli that differ only in speed), determined by well-established facts about the mean frequency of CT activity in microneurography studies (Vallbo et al., 1993, 1999; Löken et al., 2009), by clinico-anatomical and behavioural evidence by the peripheral neuropathies the reviewer mentions (Olausson et al., 2002, 2008), two neuromodulation studies using rTMS (Case et al., 2016, 2017), and several functional neuroimaging studies (Morrison et al., 2011; McGlone et al., 2012) and also by psychophysiological studies to approximate ‘CT pleasantness sensitivity’ (Crucianelli et al., 2013, 2018; Gentsch et al., 2015; von Mohr et al., 2017; Kirsch et al., 2018) and hence we wish to maintain this term, albeit with now a more explicit acknowledgement that this is an inference on our part based on prior convergent evidence.

Following the spirit of the reviewer’s suggestion, in the Discussion section, we also now explicitly acknowledge the additional controls that future studies could implement to substantiate our claims.

Introduction. “This touch on the forearm stimulates both Aβ and CT fibers; one cannot stimulate one type of fiber without stimulating the other simultaneously (except in patients without Aβ afferents, as studied by Olausson et al., 2002, 2008). […] Specifically, Aβ fiber activation is known to linearly increase with increases in velocity, while the mean frequency firing rate of CT fibers follows an inverted U shape with higher firing being in the 1-10cm/s range, and have been shown to be the only unit types for which firing patterns correlate with average psychophysiological ratings, i.e. pleasantness (Löken et al., 2009).”

Results and Discussion section. “The present study has considered CT pleasantness sensitivity as the difference between the pleasantness of CT-optimal slow touch (3cm/s) and CT-suboptimal fast touch (18cm/s). […] Moreover, the specificity of the present findings to CT fibers should be further investigated by comparing tactile stimulation on hairy (e.g. forearm) vs. glabrous skin (e.g. palm, that do not contain any CT fibers).”

2) The papers cited relating pleasantness to optimal CT stimulation (e.g. Gentsch, 2015) were undertaken in non-clinical individuals. Patients may not show the same profile of stroking rate / pleasantness judgment, e.g. may feel pleasantness at a different rate of stroking but it is assumed that the normative relationship holds.

We thank reviewer 1 for this comment that allows us to clarify the benefits of our neuropsychological/lesion approach, which does not follow the older logic of comparing lesions between a diagnostically-defined and a healthy group, but is a more advanced method using continuous measures in a single sample. This has many general neuropsychological advances and specifically in relation to the question the reviewer asks, we agree that patients, presumably based on their particular lesion profile, may not show the same profile of stroking rate/pleasantness judgement and this is precisely what our study is testing by a VLSM approach, i.e. we recruit a relatively large sample in which we did not expect all patients to show the same tactile profile and hence we examine the lesioned voxels that correspond to different behavioural deficit profiles.

We now clarify this in our manuscript as follows:

Introduction. “Contrary to other neuropsychological approaches that employ diagnostic, group comparisons, the VLSM method uses continuous measures in a single sample, and identifies which regions of the brain are crucial to a specific behavior (e.g. here CT pleasantness perception), without assuming that all patients show the same tactile profile.”

3) It is not actually that clear what the higher-level predictions (hence conclusions) actually were. Some experiences of touch are more neutral than other experiences of touch; while somatosensory cortex is not associated with construction / appraisal of emotional meaning, parts of insula and other regions are implicated in affective judgements, in part through association with interoceptive processes (e.g. bradycardia associated with affiliative interaction). Is the question then whether there is selective deficit in the experience of positive affect related to speed of touch in a brain-damaged patient group that can be localised to areas implicated in affect? There are multiple aspects to this question, so it ought to be clearly motivated.

We thank reviewer 1 in giving us the opportunity to further specify the more higher-level, theoretical aspects of our hypothesis in relation to some of the interesting and wider issues they raise. Our study addresses the question of whether the activation of CT afferents (a bottom-up rather than a top-down evaluative/appraisal mechanism; please see our specification of control and differential measures for the latter aspect), with its critical, primary cortical processing in the posterior insula, increases the likelihood of touch being perceived as positive. This hypothesis is distinguished from a number of other hypotheses assessed with control comparisons, now specified below. Our answer here also relates to the next three points raised by the reviewer so we will refer back to it.

Introduction: “C-tactile afferents have been shown to take a distinct ascending pathway from the periphery to the posterior insula (Olausson et al., 2002; Morrison et al., 2011), which is understood to support an early convergence of sensory and affective signals about the body that are then re-represented in the mid- and anterior insula, the proposed sites of integration of interoceptive information with other contextual information (Critchley et al., 2004; Craig, 2009; Evrard and Craig, 2015).”

Introduction.: “Given that right hemisphere and particularly right perisylvian regions have been previously associated with somatosensory and interoceptive perception (Dijkerman and de Haan, 2007; Preusser et al., 2015), we expected our patients to have, on average, reduced ratings of both touch intensity and pleasantness in comparison to healthy controls, and particularly in the contralesional left forearm. […] Moreover, this would give further substance to the hypothesis that the CT afferent pathway is a specialized system that allows individuals to distinguish a range of velocities that are likely to have social-affective relevance, for the purposes of further integration with sensory and affective information in the insula (Olausson et al., 2008; see Morrison et al., 2010 for discussion).

4) Continuing the point above, does the investigation of right hemisphere regions associated with the perceived affective impact of CT-optimal touch condense to whether affective experience is compromised by focal (insular) lesions; whether stroking sensation, rather than not discriminative touch is compromised by focal lesions, or whether the affective aspects of stroking sensation can uncoupled from non-affective intensity of such sensation. This is relevant as, for example, there are no equivalent (exteroceptive or interoceptive) control for pleasant sensation, so general or specific inferences are constrained. Affective reactivity to other external stimuli may also be an important individual difference, although medication with significant cognitive or mood side effect were excluded (it would be useful to state explicitly that this included β blockers, pregabalin, lamotrigine). Nevertheless, mood was recording and it is unclear how tactile anhedonia related to negative affectivity in the patients.

We thank reviewer 1 for this comment, and we now better specify the rationale and implications of our hypothesis, consistently with the higher-level theoretical aspects explained above. Specifically, the investigation of the specific right-hemisphere lesions associated with reduced CT pleasantness specificity was motivated by the hypothesis that the CT system has the unique functional role in picking the socio-affective value of gentle touch among the potential noise of other tactile and sensory information and conveying it to the posterior and anterior insula for further integration and affective processing. Indeed, as the reviewer notes in their following point we have controlled for more general effects of tactile anhedonia, as well as for more general tactile, mood and sensory deficits in our sample and we have now revised our manuscript to clarify these control procedures and analyses earlier in the manuscript (Introduction, Results and Discussion section).

Introduction. “In addition to the affective touch paradigm, to control for general pleasantness deficits (specific to touch), participants had to imagine being touched by pleasant (i.e. velvet) and unpleasant (i.e. sandpaper) materials and rate the associated pleasantness.

Given that right hemisphere and particularly right perisylvian regions have been previously associated with somatosensory and interoceptive perception (Dijkerman and de Haan, 2007; Preusser et al., 2015), we expected our patients to have, on average, reduced ratings of both touch intensity and pleasantness in comparison to healthy controls, and particularly in the contralesional left forearm. An overall reduced tactile pleasantness in patients (both in actual touch and imagined touch pleasantness ratings) would suggest tactile anhedonia linked to general right hemisphere lesions.”

Results and Discussion section. “Interestingly, none of the patients’ demographic characteristics or, neuropsychological deficits correlated significantly with their CT pleasantness sensitivity, including education, anxiety and depression scores, as well as memory as measured by the MOCA memory subscale, and working memory as measured by the Digit Span, all p>0.1 and all BF_10_ <1. Thus, low CT pleasantness sensitivity was not explained by other general cognitive and emotional deficits, as assessed in the present study. Moreover, there was no correlation between CT pleasantness sensitivities and tactile anhedonia on either forearm (as measure by the difference between the imagined pleasantness of pleasant and unpleasant material; r_31_=-.104, p=0.578, BF_10_=.259 for the contralesional forearm; r_36_=-.086, p=.618, BF_10_=.234, for the ipsilesional forearm), nor with tactile acuity as measured by intensity ratings.”

Materials and methods section. “(vi) no medication with significant cognitive or mood side-effects (e.g. pregabalin, lamotrigine);”

5) The hypothetical touch rating (velvet vs sandpaper) protocol could be mentioned in the Introduction as it pops out unexpectedly in the Results section. The rationale should be more explicit. (Is it a sub hypothesis about perceptive loss leading to imaginative loss for this channel of sensation?). In the Materials and methods section, it could be usefully described in more detail: were people asked to imagine velvet and sandpaper drawn across the skin at different velocities in order to mirror the effects of CT optimal / suboptimal stimulation. Would an interaction be expected?

We thank the reviewer for this suggestion, and we have now included this scope of measuring this dimension of imagined tactile pleasantness, as well as all the other control measures of our study which we think further specify the relation between our results and our hypotheses (see point 4 and additions below).

Moreover, we were not trying to match imagined touch to actual touch at an embodied level, but we aimed to ensure that patients do not have deficits in imaging or, expressing tactile pleasure beyond the level of actual tactile stimulation, which is controlled for by our main task by the faster trials.

Introduction: “In addition to the affective touch paradigm, to control for general pleasantness deficits (e.g. tactile anhedonia), participants had to imagine being touched by pleasant (i.e. velvet) and unpleasant (i.e. sandpaper) materials and rate the associated pleasantness.”

(…)

“An overall reduced tactile pleasantness in patients (both in actual touch and imagined touch pleasantness ratings), would suggest tactile anhedonia, linked to general right hemisphere lesions.”

Results and Discussion section:” Additionally, as a control for a general pleasantness deficit, patients rated how pleasant it would be to be touched by a typically pleasant material and a typically unpleasant fabric. As done for CT pleasantness sensitivity, imagined tactile pleasantness sensitivity was computed as the difference between pleasant and unpleasant materials pleasantness ratings, for each patient. We considered the same patients as for the CT pleasantness sensitivity VLSM analysis (N=36 as we had missing data for 5 of them) and ran a VLSM analysis with this top-down tactile pleasantness sensitivity as predictor. This yielded significant voxels subcortically in the caudate, thalamus, putamen and pallidum, but crucially, not the insula, suggesting that the above results are specific to applied tactile stimuli and not more general pleasantness comparisons (see Supplementary file 1)”

Materials and methods section. “No other instruction was given to the participants (neither speed nor pressure of the imagined touch).”

6) There is mention of interoception and interoceptive awareness e.g. in the Discussion section. The consensus definition of interoception is the sensation of the internal state of the body. If the authors maintain Craig's proposal that stroking and tickle sensation are honorary interoceptive sensations by virtue of having un- or poorly- myelinated fibres and sharing in part the same spinal and perhaps central pathways as viscerosensory afferents then this should be strongly and explicitly argued. If this is argued that the core aspect is that these sensations are affective/emotive then the issue about prediction / hypothesis underlying the experimental design is relevant. The term interceptive awareness has evolved since Garfinkel, 2013 in the light of consciousness science equating awareness with metacognitive insight, interoceptive sensibility being one term used for reported perception or misperception of visceral sensation.

We thank reviewer 1, giving us the opportunity to specify our view. As mentioned in previous comments we made several additions in the introduction and in the discussion to clarify our definitions of interoception, and justification of including affective touch as an interoceptive modality.

Moreover, we have changed our terminology, to be more accurate and in line with recent findings (from interoceptive awareness, that refers more to confidence in judgements, to interoceptive perception, that captures more sensing the body sensations; Garfinkel et al., 2016; Critchley and Garfinkel, 2017).

Results and Discussion section: “Taken together, our findings support previous findings about the functional organization and role of the human insula(Craig, 2010; Cauda et al., 2011; Kurth et al., 2010; Heydrich and Blanke, 2013; Ronchi et al., 2015; Salomon et al., 2018; see review by Evrard, 2019, on recent findings on the organization of the insula in non-human primates), consisting of specialized substrates organized in a posterior to anterior structural progression, with posterior parts representing the primary cortical representations of interoceptive stimuli from contralateral body parts and more anterior parts, tested here in the right hemisphere, acting as integration areas for sensory signals and contextual cues ultimately leading to interoception. […] As VLSM methods preclude direct comparison between CT pleasantness sensitivity deficits on the contralesional and ipsilesional forearm at the brain level; future studies should investigate further the posterior-anterior insula segregation in relation to affective touch as an interoceptive modality.”

7) Anatomical understanding of insula has been enhanced greatly through the work of Evrard. This could be reasonably cited.

We thank reviewer 1 for the advice; we have now included reference to Evrard work in the Introduction and Results and Discussion section (Results and Discussion section Evrard 2019; Introduction Evrard and Craig, 2015).

Results and Discussion section. “Taken together, our findings support previous findings about the functional organization and role of the human insula (Craig, 2010; Cauda et al., 2011; Kurth et al., 2010; Heydrich and Blanke, 2013; Ronchi et al., 2015; Salomon et al., 2018; see review by Evrard, 2019 on recent findings on the organization of the insula in non-human primates), consisting of specialized substrates organized in a posterior to anterior structural progression, with posterior parts representing the primary cortical representations of interoceptive stimuli from contralateral body parts and more anterior parts, tested here in the right hemisphere, acting as integration areas for sensory signals and contextual cues ultimately leading to interoception.”

Reviewer #2:The manuscript describes a lesion study in 59 patients investigating the impact of insular damage on CT functioning. The study is very interesting, very timely and of potential high contribution for the field. The methods are sound. The Introduction, Discussion section and Results section are quite short. This could be a benefit, but it makes the paper hard to understand at several places.My main concern is the mismatch between hypothesis, result and discussion. The authors hypothesize the pleasantness difference (CT optimal- suboptimal velocities) to be higher when patients are contralateral stroked than in other conditions (healthy controls or patients ipsilateral), This would be a 3-way interaction (velocity by group by side) which is not tested. In addition, Figure 1 does not support this hypothesis. Anyhow, the authors write in the Discussion section, that the hypothesis is supported.Support for the hypothesis comes from the VLSM approach. I needed to reread the results several times in order to understand this. Especially as the methods are described after the results, it is unclear what the authors actually did. It would be much easier, if the authors would introduce the approach and the corresponding hypothesis in the Introduction.

We thank reviewer 2 for these positive comments and giving us the opportunity to improve our manuscript. We have restructured the manuscript along your comments and reviewer 1’ comments and hope it to be clearer. Please also note the strict word limit of the journal for Short Report.

Indeed, as the reviewer notes the operationalization of our hypothesis and the chosen method is a VLSM study where the interaction the reviewer mentions does not apply. VLSM (Voxel-based, Lesion-Symptom Mapping) studies differentiate patients in a single sample based on the voxels that are damaged in the sub-set of patients that have a behavioural deficits versus those that do not. We apologize for any lack of clarity on our behalf and we have now specified our approach and hypotheses in the introduction, also in response to reviewer 1 (points 1 to 3 in particular). As we say in the manuscript, we are not expecting such a general effect of the stroke on CT pleasantness sensitivity in our patients, but rather we were looking for the specific lesion-symptom relation, which is optimally examined by a VLSM methodology.

Introduction “Contrary to other neuropsychological approaches that employ diagnostic, group comparisons, the VLSM method uses continuous measures in a single sample, and identifies which regions of the brain are crucial to a specific behavior (e.g. here CT pleasantness perception), without assuming that all patients show the same tactile profile.”

Introduction. “An overall reduced tactile pleasantness in patients (both in actual touch and imagined touch pleasantness ratings) would suggest tactile anhedonia linked to general right hemisphere lesions. […] Moreover, this would give further substance to the hypothesis that CT afferent pathway is a specialized system that allows individuals to distinguish a range of velocities likely to have social-affective relevance, for the purposes of further integration with sensory and affective information in the insula (Olausson et al., 2008; see Morrison et al., 2010 for discussion).”

Results and Discussion section. “First, we investigated the effect of right hemisphere lesions on the perception of touch intensity and pleasantness, on the contralesional and ipsilesional forearm separately, by comparing stroke patients’ and healthy controls’ intensity and pleasantness ratings in turn. In line with the high percentage of contralesional tactile deficits in right hemisphere stroke patients (including in our patients’ sample, see Materials and methods section), patients, as compared to healthy controls, perceived touch, regardless of velocity, as less intense on the contralesional forearm.”

Results and Discussion section. “However, no interaction between touch type and group was found (contralesional: F(1,53)=0.393, p=0.533, ηp2 = .007, BF_10_=0.371, Figure 1C; ipsilesional: F(1,59)=0.073, p=0.788, ηp2 = .001, BF_10_=0.287, Figure 1D; hypothetical imagined tactile touch pleasantness: F(1,70)=.061, p=0.806, ηp2=.001, BF_10_=0.270), suggesting that right hemisphere lesions in general do not necessarily lead to reduced CT pleasantness sensitivity, and confirming that any differential deficits in the pleasantness perception of CT-optimal versus CT-suboptimal touches at the individual level would relate to specific lesions rather than general stroke effects.”

Results and Discussion section. “The present study aimed to investigate the lesion patterns and neuropsychological deficits that may be associated with the inability of certain stroke patients to distinguish the pleasantness of CT-optimal versus CT-suboptimal touches. Accordingly, CT pleasantness sensitivity was calculated as the difference between the pleasantness of CT-optimal and CT-suboptimal touches. As a convention, CT pleasantness sensitivity inferior or equal to zero is considered as low in CT pleasantness sensitivity, i.e. low CT affective touch perception (Crucianelli et al., 2018).”

Results and Discussion section: what kind of patients are the ones with intact sensation? How did you measure this?

We thank reviewer 2 for this comment, we have now clarified this in the manuscript. Patients with intact sensations are a subsample of the 59 patients, patients who rated all touch trials above 2 on the intensity scale (i.e. could feel all the trials).

Introduction. “This was also the case when considering only patients that had intact tactile perception on the contralesional forearm, i.e. could feel all the touch trials (N=25, F(1,43)=9.880, p=0.003, ηp2 = .187, Figure 1—figure supplement 1; see Materials and methods section for details).”

Materials and methods section: Moreover, as supplementary analyses, we also considered patients with intact tactile perception on the contralesional forearm. For these analyses, only patients that gave intensity ratings above 2 were included (N=25).

In the Results and Discussion section, the authors introduce a comparison between hypothetical velvet stroking and sandpaper stroking. This is very surprising and could also be introduced better in the introduction part. I would actually remove this part, which is in my eyes unnecessary for the hypothesis but confusing for the reader.

We thank reviewer 2 for this comment. We agree that this analysis was not well introduced and justified. As in response to reviewer 1 who wanted more precision on this analysis, we have now clarified and expanded it. We hope it is less confusing.

Introduction. “In addition to the affective touch paradigm, to control for general pleasantness deficits (specific to touch), participants had to imagine being touch by pleasant (i.e. velvet) and unpleasant (i.e. sandpaper) materials and rate the associated pleasantness.”

Results and Discussion section. “A similar general tactile anhedonia (reduced pleasantness ratings) was observed in our patients as compared to the controls for imagined tactile pleasantness, when patients had to rate how pleasant it would be to be touched by pleasant and unpleasant fabric (F(1,57) = 55.918, p<0.001, ηp2=.495, Figure 1—figure supplement 2).”

The generally reduced pleasantness scores in lesion patients are most likely due to the enhanced depression and reduced attention in this group, a confounder analysis would clarify this.

We thank reviewer 2 for this comment, as for reviewer 1’s comment 4, we have run correlations between depression scores (as measured by the HADS), memory deficits (as measured by the MOCA memory subscale) and averaged pleasantness ratings; this yielded non-significant correlations (all p-values>0.50, BF_10_<1). Moreover, as explained to reviewer 2, given non-significant correlations with CT pleasantness sensitivities and neuropsychological tests, these variables were not included in the VLSM analyses as nuisance covariates (as advised by de Haan and Karnath, 2018)

Results and Discussion section. “Interestingly, none of the patients’ demographic characteristics or, neuropsychological deficits correlated significantly with their CT pleasantness sensitivity, including education, anxiety and depression scores, as well as memory as measured by the MOCA memory subscale, and working memory as measured by the Digit Span, all p>0.1 and all BF_10_ <1. Thus, low CT pleasantness sensitivity was not explained by other general cognitive and emotional deficits, as assessed in the present study. Moreover, there was no correlation between CT pleasantness sensitivities and tactile anhedonia on either forearm (as measure by the difference between the imagined pleasantness of pleasant and unpleasant fabric; r_31_=-.104, p=0.578, BF_10_=.259 for the contralesional forearm; r_36_=-.086, p=.618, BF_10_=.234 for the ipsilesional forearm), nor with tactile acuity as measured by intensity ratings.”

[Editors' note: further revisions were suggested prior to acceptance, as described below.]Reviewer #2:The revised manuscript improved in terms of understandability and the authors clearly put effort in the rebuttal. However, my concerns are not eliminated.I still have some questions about the "pleasantness sensitivity"

We thank the reviewer for their careful comments however we wish to stress that such comments seem to correspond to and thus perhaps be motivated by a different, older methodological approach to lesion studies. Namely, one that relies on comparisons between diagnosis-based clinical groups and/or between clinical groups and healthy controls. Please note that as we explained in the text, in the previous revision and our points below, the VLSM method is not based on such comparisons, i.e. it does not use patients and controls to be grouped by either lesion, diagnosis or behaviour, but rather makes use of continuous behavioural and lesion information (e.g. Bates et al., 2003) and this is what we have applied to our relatively, large sample of right hemisphere stroke patients.

- It is not clear to me why the authors used 18cm/s as a control for suboptimal CT activation. The microneurography studies are far from clear for this velocity and 30cm/s would be the standard with supporting microneurography data.

We use the 18cm/s velocity on the basis of both behavioural reasons and microneurography indications:

a) Behavioural: While we agree with the reviewer that several studies use 30cm/s as a control velocity, this psychophysical choice is not without limitations in behavioural studies for a number of reasons including (1) the very poor ecological validity of applying this kind of speed on human skin in comparison to 3cm/s and 18cm/s (2) the very different amount of strokes or time required to administer this kind of touch on the same skin area in comparison to 3cm/s and to 18cm/s, (3) the related differences in the kind of emotions and mental states people consciously associate with increasing velocities, already present at 18cm/s (Kirsch et al., 2018), (4) the very different attentional demands this kind of velocity generates in comparison to 3cm/s and 18cm/s and (5) the related difficulties in administering such stimuli manually and by the bedside in right-hemisphere patients with known attentional, arousal and visuospatial difficulties. The above reasons may be of lesser relevance in microneurography studies, but they are of great importance in behavioural and clinical studies. Thus, our lab has for the past 6 years conducted more than 12 published studies using the 18cm/s velocity as a CT non-optimal velocity, all of which showed that this velocity is subjectively perceived as significantly less pleasant (e.g. Crucianelli et al., 2013; Gentsch et al., 2015; Kirsch et al., 2018; von Mohr, Kirsch and Fotopoulou, 2019) and it has different, indirect, implicit effects on behavioural and cognition (e.g. von Mohr et al., 2017; Krahé et al., 2016; 2018; Panagiotopoulou et al., 2017; 2018; Crucianelli et al., 2018) than the 3cm/s velocity.

b) Microneurography studies have indeed tested 3cm/s versus the 30cm/s velocity. However, they have also tested intermediate velocities (i.e. 9cm/s and 10cm/s) and they have consistently found that these generate both CT and subjective responses that fall somewhere between the 3cm/s and the 30cm/s (see Loken et al., 2009; and Ackerley et al., 2014). Hence, the upper component of the inverted U shape and hence to our knowledge, no studies has ever considered any velocity faster than 10cm/s as optimal CT velocity. Our choice of a sub-optimal velocity is thus well above the accepted optimal range of CT fibers.

In sum, taking both behavioural and microneurography considerations and evidence into account we elected 18cm/s as the best control velocity for our paradigm.

- How did the authors deal with those cases where roof or bottom effects were observed (e.g. rating all touch highly positive or negative)?

Our data did not show ceiling or floor effects of the kind the reviewer is concerned about (only 1 patient out of the 59 rated both touch velocities at the maximum and none of the patients and healthy controls rated the touch as not pleasant at all). In all cases, our continuous differential measure, motivated by our specific hypothesis, is designed to indeed take this into account, i.e. touch insensitive ceiling or floor responses would lead to a low sensitivity, i.e. a very small differential.

- How many of the healthy controls have a "negative pleasantness sensitivity"? And how many do not seem to differentiate between 3 and 18cm/s (no difference or very minor difference in pleasantness ratings)?

Individual differences are only meaningful in the patient group and it is these differences only that are taken into account in our lesion analyses. Any comparison involving the smaller group of healthy participants for behavioural validity purposes is meaningful only at the group level and it does not relate to our lesion analysis. For the reviewers’ interest, we note here that for the contralesional arm, one out of 20 healthy participants rated both velocities as similar in pleasantness and two others found the 18cm/s velocity slightly more pleasant than the optimal one.

I may have missed it, but I cannot find the place where the authors show that the construct they call CT pleasantness sensitivity differs between patients with posterior insula lesions and healthy controls. Instead, the authors report differences between patients with various lesions. This does not relate to the abstract or to the hypothesis.

We apologize if our response in the previous round of revision was not clear. We had changed both the abstract and our hypotheses to be clearly about lesion-patients and not about healthy controls vs. patients. Our aim is not to compare patients with posterior insula lesions to healthy controls, but to find which region is crucial to CT pleasantness perception. For this reason, we used a continuous VLSM approach in patients with right hemisphere lesion.

Indeed, as written in the Introduction “Contrary to other neuropsychological approaches that employ diagnostic, group comparisons, the VLSM method uses continuous measures in a single sample, and identifies which regions of the brain are crucial to a specific behavior (e.g. here CT pleasantness perception), without assuming that all patients show the same tactile profile.”. For this reason, the VLSM method do not need to compare healthy controls and patients, and in fact looking whether CT pleasantness sensitivity differs between patients with posterior insula lesions and healthy controls would be double dipping of the data, and make the analyses circular. For this reason, we would like to argue that it would not be scientifically sound to do so.

Note: Below is how we introduced our aim and hypotheses:

Abstract: " We report the first human lesion study on the perception of C-tactile touch in right hemisphere stroke patients (N = 59), revealing that right posterior and anterior insula lesions reduce tactile, contralateral and ipsilateral pleasantness sensitivity, respectively.

Aim (Introduction): "Accordingly, we aimed to investigate for the first time the right hemisphere regions which are necessary for the perceived affectivity of CT-optimal touch, applying a voxel-based lesion symptom mapping approach (VLSM; Bates et al., 2003) in a large, consecutively recruited cohort of patients (N=59) with recent, first-ever, right hemisphere lesions following a stroke.

Hypotheses (Introduction): "Crucially, given the assumed neurophysiological specificity of the CT system, we expected that more specific lesions to the posterior insula (Morrison, 2016) would reduce the affective sensitivity of these patients to CT-optimal touch, over and above general effects of anhedonia, tactile acuity and other neuropsychological deficits caused by the broader lesion profile of our whole sample. In other words, an intact posterior insula should be necessary for the added affective sensitivity that the CT fibers are conveying during touch optimally activating the CT system versus an identical touch and social context that does not activate this afferent pathway optimally."

I don't understand why the authors did not control for potential sources of variance in the VTLM. The authors argue that "However, as explained in point 4 above, no correlations were found between CT pleasantness sensitivity (or pleasantness ratings) and any of the neuropsychological tests that differed between healthy controls and right hemisphere stroke patients (HADS Depression scale, Digit Span backward, MOCA and Premorbid IQ_WTAR). " However, the main observation the authors report is not between healthy controls and patients, but within groups of patients.

We apologize if our answer was not clear, it might have been confusing as we were also referring to our response to reviewer’1 comment. Indeed, we are focusing mainly on patients as a continuous group in the present study.

In accordance with recent recommendations on how to conduct lesion-behaviour mapping by de Haan and Karnath, (2018), it is advised to include only nuisance covariates in the VLSM analyses, such as the severity of frequently co-occurring deficits, that may correlate with the cognitive function of interest. For this reason, we first determined whether variables that differed between right hemisphere stroke patients and healthy controls are nuisance variable worth including by looking if there is a correlation between the variable of interest and the nuisance variable. If no significant correlation is found (within the patient group), the variable does not need to be included – and including it would reduce power unnecessarily.

As written in our previous rebuttal, given non-significant correlations between CT pleasantness sensitivities and neuropsychological tests in patients (and not considering healthy controls here), these variables were not included in the VLSM analyses as nuisance covariates (as advised by de Haan and Karnath, 2018).

More precisely:

Results and Discussion section. “Interestingly, none of the patients’ demographic characteristics or, neuropsychological deficits correlated significantly with their CT pleasantness sensitivity, including education, anxiety and depression scores, as well as memory as measured by the MOCA memory subscale, and working memory as measured by the Digit Span, all p>0.1 and all BF_10_ <1.”